# In Vitro Anti-*Candida* Activity and Action Mode of Benzoxazole Derivatives

**DOI:** 10.3390/molecules26165008

**Published:** 2021-08-18

**Authors:** Monika Staniszewska, Łukasz Kuryk, Aleksander Gryciuk, Joanna Kawalec, Marta Rogalska, Joanna Baran, Edyta Łukowska-Chojnacka, Anna Kowalkowska

**Affiliations:** 1Centre for Advanced Materials and Technologies CEZAMAT, Warsaw University of Technology, Poleczki 19, 02-822 Warsaw, Poland; jbaran@ch.pw.edu.pl; 2Department of Virology, National Institute of Public Health-NIH-National Research Institute, Chocimska 24, 00-791 Warsaw, Poland; lkuryk@pzh.gov.pl; 3Clinical Science, Targovax Oy, Saukonpaadenranta 2, 00180 Helsinki, Finland; 4Faculty of Chemistry, Warsaw University of Technology, Noakowskiego St. 3, 00-664 Warsaw, Poland; alek.gryciuk@gmail.com (A.G.); joanna.kawalec.stud@pw.edu.pl (J.K.); martarogalska98@gazeta.pl (M.R.); elukowska@ch.pw.edu.pl (E.Ł.-C.)

**Keywords:** benzoxazole, *N*-phenacyl, *Candida* spp., action mode, in vitro

## Abstract

A newly synthetized series of *N*-phenacyl derivatives of 2-mercaptobenzoxazole, including analogues of 5-bromo- and 5,7-dibromobenzoxazole, were screened against *Candida* strains and the action mechanism was evaluated. 2-(1,3-benzoxazol-2-ylsulfanyl)-1-(4-bromophenyl)ethanone (**5d**), 2-(1,3-benzoxazol-2-ylsulfanyl)-1-(2,3,4-trichloro-phenyl)ethanone (**5i**), 2-(1,3-benzoxazol-2-ylsulfanyl)-1-(2,4,6-trichlorophenyl)ethanone (**5k**) and 2-[(5-bromo-1,3-benzoxazol-2-yl)sulfanyl]-1-phenylethanone (**6a**) showed anti-*C. albicans* SC5314 activity, where **5d** displayed MIC_T_ = 16 µg/mL (%R = 100) and a weak anti-proliferative activity against the clinical strains: *C. albicans* resistant to azoles (Itr and Flu) and *C. glabrata*. Derivatives **5k** and **6a** displayed MIC_P_ = 16 µg/mL and %R = 64.2 ± 10.6, %R = 88.0 ± 9.7, respectively, against the *C. albicans* isolate. Derivative **5i** was the most active against *C. glabrata* (%R = 53.0 ± 3.5 at 16 µg/mL). Benzoxazoles displayed no MIC against *C. glabrata*. Benzoxazoles showed a pleiotropic action mode: (**1**) the total sterols content was perturbed; (**2**) 2-(1,3-benzoxazol-2-ylsulfanyl)-1-(3,4-dichlorophenyl)ethanol and 2-(1,3-benzoxazol-2-ylsulfanyl)-1-(2,3,4-trichlorophenyl)ethanol (**8h**–**i**) at the lowest fungistatic conc. inhibited the efflux of the Rho123 tracker during the membrane transport process; (**3**) mitochondrial respiration was affected/inhibited by the benzoxazoles: 2-(1,3-benzoxazol-2-ylsulfanyl)-1-(4-chlorophenyl)ethanol and 2-(1,3-benzoxazol-2-ylsulfanyl)-1-(4-bromophenyl)ethanol **8c**–**d** and **8i**. Benzoxazoles showed comparable activity to commercially available azoles due to (**1**) the interaction with exogenous ergosterol, (**2**) endogenous ergosterol synthesis blocking as well as (**3**) membrane permeabilizing properties typical of AmB. Benzoxazoles display a broad spectrum of anti-*Candida* activity and action mode towards the membrane without cross-resistance with AmB; furthermore, they are safe to mammals.

## 1. Introduction

Fungal diseases represent a critical worldwide health problem and they are one of the main causes of morbidity and mortality. Moreover, only a few classes of antifungals are currently available to treat patients with fungal infections due to their high toxicity [1]. Therefore, it is still relevant to develop new alternative compounds, without harmful side-effects for patients, to prevent the emergence of fungal resistance. We, thus, undertook in vitro studies of anti-*Candida’s* potential of new benzoxazoles. The benzoxazole ring is an important pharmacophore, that has for years been used to synthesize new biologically active compounds, including antifungal ones [2,3,4,5,6,7,8]. Among these compounds, there are derivatives of benzoxazoles active against *Candida* spp. [4,5,6,7,8,9,10,11,12]. Taking into account the structure of the compounds: **5**–**8**, the confirmed activity of the 2-mercaptobenzoxazole derivatives against *Candida* spp. is particularly important [13,14]. On the other hand, the compounds *S*-functionalized with the phenacyl or halogen-phenacyl group also exhibit antifungal activity [14,15,16,17,18,19,20].

This prompted us to synthesize the *N*-phenacyl derivatives of 2-mercaptobenzoxazole, including analogues of 5-bromo- and 5,7-dibromobenzoxazoles. The ketones were reduced to respective alcohols. In the study, we evaluated the toxicity of various benzoxazoles towards the mammalian cell lines as well as the fungistatic and fungicidal effect against the reference and clinical isolates resistant to azoles. The exogenous ergosterol affinity of the most active and randomly selected benzoxazoles was tested. Benzoxazoles inhibitory activity towards ergosterol synthesis was studied using the biochemical (high-performance liquid chromatography HPLC) and spectrophotometric method (SPE). Benzoxazole permeability/lytic activity against the fungal membrane was assessed via flow cytometry. Benzoxazoles exhibit the candidacidal mechanism via accidental cell death (ACD).

## 2. Results

### 2.1. Synthesis of Benzoxazoles

The synthesis of benzoxazole derivatives **5**–**8** is outlined in Scheme 1 and Scheme 2, as well as benzoxazoles summarized in Table 1. At first, 5-bromo-2-mercaptobenzoxazole **2** and 5,7-dibromo-2-mercaptobenzoxazole **3** were obtained according to procedures reported earlier [21] with the modification of one step. Instead of using a mixture of stannous chloride dihydrate and concentrated hydrochloric acid in a methanol reduction in the respective nitrophenols [21], it was carried out by NaBH_4_ in the presence of activated carbon in the THF-H_2_O mixture [22]. Ketones **5**–**7** were synthesized from 2-mercaptobenzoxazoles **1**–**3** and phenacyl bromides or chlorides **4** by *S*-alkylation carried out in the K_2_CO_2_/MeCN system (Scheme 1). Derivatives of 2-mercaptobenzoxazole **5** were easily obtained in high yields, while in the case of bromo-derivatives **6**–**7**, more by-products were formed, so their yields were lower.

Reduction in ketones **5** with NaBH_4_ in MeOH in room temperature afforded the mixture of products, alcohols **8** and respective 2-hydroxybenzoxazoles [23]. Decreasing of the reduction temperature to 0–3 °C allowed to obtain compounds **8** with moderate to good yields (Scheme 2). All alcohols **8** were isolated, but some of them turned out to be unstable not only at room temperature, but even when stored in the refrigerator (ca 3–5 °C). Additionally, attempts to obtain respective alcohols, derivatives of bromobenzoxazoles **9** and **10**, by the reduction in ketones **6 and 7** were rather unsuccessful, because of the formation of complicated mixtures of products. All alcohols **9** and **10**, isolated in low yields and moderate purity, were unstable so they were not fully characterized and biologically tested.

All new stable compounds were fully characterized by ^1^H NMR, ^13^C NMR, IR and HRMS analyses.

**Table 1 molecules-26-05008-t001:** Yields of ketones **5**, **6**, **7** and alcohols **8**.

Lp.	Benzoxazole	4, ArCOCH_2_X ^a^	Ketone (%)		Alcohol (%)	
1	**1**	C_6_H_5_	**5a** [24,25,26,27]	98	**8a**	− ^b^
2	**1**	4-FC_6_H_4_	**5b**	91	**8b**	88
3	**1**	4-ClC_6_H_4_	**5c** [24]	95	**8c**	91
4	**1**	4-BrC_6_H_4_	**5d** [24,27]	96	**8d**	95
5	**1**	2,4-F_2_C_6_H_3_	**5e**	88	**8e**	57
6	**1**	2,4-Cl_2_C_6_H_3_	**5f**	87	**8f**	61
7	**1**	2,5-Cl_2_C_6_H_3_	**5g**	91	**8g**	37 ^c^
8	**1**	3,4-Cl_2_C_6_H_3_	**5h**	67	**8h**	73
9	**1**	2,3,4-Cl_3_C_6_H_2_	**5i**	92	**8i**	87
10	**1**	2,4,5-Cl_3_C_6_H_2_	**5j**	91	**8j**	89
11	**1**	2,4,6-Cl_3_C_6_H_2_	**5k**	64	**8k**	− ^d^
12	**2**	C_6_H_5_	**6a**	71	**9a**	39 ^c,e^
13	**2**	4-FC_6_H_4_	**6b**	55	**9b**	36 ^c,e^
14	**3**	C_6_H_5_	**7a**	53	**10a**	33 ^c,e^
15	**3**	4-FC_6_H_4_	**7b**	39	**10b**	− ^e^

^(a)^ Phenacyl bromide (Ar = C_6_H_5_, X = Br) and 4-chlorophenacyl bromide (Ar = 4-ClC_6_H_4_, X = Br) were used; for all other compounds **4**, X = Cl. ^(b)^ Decomposition before purification. ^(c)^ Compound unstable. ^(d)^ Compound containing some by-products. ^(e)^ The mixture impossible to separate.

### 2.2. Anti-Candida Activity of Benzoxazoles

After the appropriate 48 h incubation time, the presence of growth was assessed visually and using the spectrophotometric method (SPE). Derivatives: **5d**, **5i, 5k** and **6a** showed anti-*C. albicans* SC5314 ATCC activity (Table 2, Appendix A), where **5d** displayed the total visual MIC_T_ after 48 h. Cell growth inhibition (SPE end-point %R) and MIC were also determined (Table 3, Appendix A) against the *Candida* clinical isolates. Derivatives **5k** or **6a** at 16 µg/mL showed the MIC_P_ partial values and, respectively, cell inhibition app. at % = 64.2 ± 10.6 or % = 88.0 ± 9.7 against the *C. albicans* isolates (Table 2). Among the tested benzoxazoles, **5i** was the most active at 16 µg/mL against *C. glabrata* (% = 53.0 ± 3.5 in Table 3 and % = 36.9 ± 4.1 in Appendix A)**.** Benzoxazole’s MIC was not observed against the *C. glabrata* isolate.

### 2.3. Assessment of the Cytotoxicity against the Mammalian Cell Lines

In the study, the primary cell lines suitable and routinely employed for the compound cytotoxic examination were tested [29,30]. These lines were temporarily available in our laboratory and they were chosen due to the most sensitive models used for assessing the toxicity of compounds [29,30].

Among **5d**, **6a** and **7a** tested against the Vero cells, a moderate cytotoxic effect (≤63.8% ± 8.7) was observed at 256 µg/mL (Table 4 and Appendix A). In the case of these compounds, a weak or lack of cytotoxicity (app. at 18.4% ± 0.4 to 2.1 ± 0.1) at the conc. ranged from 128 to 1 µg/mL against Vero was noted. Derivatives **5a** or **5j** at 512 µg/mL showed a cytotoxicity, respectively, at 40.64% ± 3.0 or 50.75% ± 0.0. Contrariwise, **8h**, **8c**, **8i** and **8d** at the range of conc. starting from 512 to 128 µg/mL displayed a strong cytotoxicity (84.38% ± 0.1–63.26% ± 0.3) against MRC-5 (Table 4 and Appendix A). To sum up, the tested benzoxazoles at the antifungal concentrations showed a lack of cytotoxicity against the mammalian cells.

### 2.4. Estimation of the Ergosterol Content in the Benzoxazole-Treated Candida Blastoconidia

The results of the spectrophotometric (SPE) analyses showed a conc.-dependent decrease in the ergosterol content when the *C. albicans* ref. strain was grown, respectively, at conc. of 16 or 4 µg/mL of **5d** (Table 5). In comparison to the untreated control, the mean of the total ergosterol content in the **5d**-treated *C. albicans* ref. strain decreased app. by 1.3-fold at 16 µg/mL and 1.2-fold at 4 µg/mL. Derivative **5d** displayed the reverse action mode against the *C. albicans* isolate, e.g., 1.3-fold at 16 µg/mL and 1.7-fold increase at 4 µg/mL. Exposure of the ref. cells to **6a** or **7a** generated a higher ergosterol content at 16 µg/mL with a mean decrease in the total of its content, app. by one-fold (**6a** and **7a**) for the ref. strain vs. the untreated control. The **6a**- or **7a**-treated clinical isolate displayed the mean decrease app. by 1-fold (**6a** and **7a**) at conc. of 16 µg/mL and by 1.5-fold at 4 µg/mL, compared to the untreated control. Using the SPE analyses, we showed that the differences in the ergosterol content between the benzoxazole-treated vs. untreated control were statistically significant (*p* = 0.003).

Moreover, the ergosterol content was measured using HPLC (Table 6 and Appendix A). Interestingly, when applying **5d** at 16 µg/mL (MIC_T_), the treatment favoured the increase in the ergosterol content app. by 0.9-fold vs. the untreated control, while 4 µg/mL inhibited the ergosterol content app. by 1.3-fold vs. the control (Table 6). Contrariwise, we observed a 1.5- or 1.3-fold-decreased ergosterol content, respectively, for **6a** or **7a** at 16 µg/mL vs. the untreated control. The content of ergosterol in the ref. strain was reduced in the range of 1.2-1.5-fold for all the tested compounds at 4 µg/mL. In the case of the clinical isolate, **7a** at conc. of 16 µg/mL induced 0.9-fold higher ergosterol vs. the untreated control. On the contrary, **5d** or **6a** at 16 µg/mL reduced the ergosterol content by 1.9- or 3-fold, respectively, vs. the untreated control. These compounds at 4 µg/mL reduced the content of ergosterol by the following levels: 1-fold (**5d**), 2-fold (**6a**) or 1.4 (**7a**). Using HPLC, we showed the statistically significant differences in the ergosterol content between the benzoxazole-treated and the untreated control (*p* = 0.007).

Based on the method comparison studies (Regression and Pearson correlation coefficient), it was shown that there was a large bias in these two methods used to assess the ergosterol content. The Pearson correlation coefficient (r = 0.5) and *p* = 0.12 indicated a moderate correlation between these two methods. The SPE method produced slightly more accruable results due to the smaller *p*-value. In spite of this, there was *p* < 0.05 for these two methods; thus, both were worth applying.

### 2.5. Cell Death Assessment Using Flow Cytometry

To assess the ability of the benzoxazole derivatives to induce accidental cell death (ACD) in *C. albicans*, the blastoconidial cells were treated with compounds at 16 µg/mL (Figure 1) or 4 µg/mL (Appendix A) and % of the cell death was assessed by the appearance of propidium iodide (PI) staining using flow cytometry. We found that the treatment of the cells with benzoxazoles led to an irreversible plasma membrane breakdown (app. ~99.52%) vs. the untreated blastoconidial control (99.77%). Additionally, benzoxazoles at 16 or 4 µg/mL induced necrosis in the *C. albicans* protoplasts.

### 2.6. Efflux Study in the Benzoxazole-Treated C. albicans Cells

The least active fungistatic compounds were tested for the Rho123 efflux at the highest (160 or 16 µg/mL) or the lowest (1.25 or 0.125 µg/mL) antifungal conc. tested (Figure 2, Appendix A). The SPE reading revealed that **5j** and **8h** at 0.125 µg/mL as well as **5a** and **5e** at 16 µg/mL increased the efflux vs. the untreated control (OD_600nm_ = 3901). Generally, the fungistatic compounds tested at lower conc. induced the cell efflux activity. In details, when all compounds at two different conc. were tested, there were differences in the efflux as follows: **5j** (1-fold increase in the efflux at the lowest conc. tested vs. the highest one); **8d** (1.2-fold); **5a** and **8c** (1.6-fold); **5e** (1.7-fold). On the contrary, **8****i** caused a decrease in the efflux (4.6-fold) at the lowest conc. tested vs. the highest one. Although all benzoxazoles exhibited a comparable fungistatic effect, they showed a diversity in the cell efflux inhibition. Derivatives: **8c**, **8i** and **8d** inhibited the blastoconidial efflux. The Rho123 efflux in the benzoxazole-treated SC5314 was stimulated by the glucose addition, but since it gradually decreased along with the increasing concentration of **8d**, this derivative was taken as an illustrative example.

### 2.7. Determination of Antifungal Activity of Benzoxazole Derivatives in the Presence of Exogenous Ergosterol

To address whether exogenous ergosterol could relieve benzoxazole stress in *C. albicans*, blastoconidia at log phase were harvested and then treated with benzoxazoles at various concentrations in a medium containing ergosterol at 20 mM (Figure 3 and Appendix A). The itraconazole-treated blastoconidia in the medium supplemented with ergosterol and blastoconidia growing in the medium containing ergosterol were used as the controls. After 96 h of **8d** or **8f** treatment, the blastoconidial cell survival slightly decreased at the highest concentrations of compounds. The activity of **8d** and **8f** was comparable to Itr. Thus, the results were consistent with the antifungal activity of Itr, showing that exogenous ergosterol reduced the *C. albicans* tolerance to **8d** and **8f**. Differences in the antifungal effect between Itr and **5e** as well as Itr vs. **8h** were noted.

As it was shown in Appendix A, benzoxazoles **5a**, **5j**, **8c** and **8i** displayed no interaction with ergosterol as the curves differed from the Itr action. The curves of **5a**, **5j**, **8c** and **8i** differed from Itr vs. the growth control curves.

### 2.8. Anti-Candida Effect of Benzoxazole Derivatives Combined with Amphotericin B (AmB). FIC Index Calculation

As it was shown in Table 7, we summarized the combined activities of **5d** with amphotericin (AmB) in the in vitro checkboard interactions against *C. glabrata*. The FIC index interpretations for the activities of **5d** with AmB against *C. glabrata* recorded an indifference for all combinations. Synergism or antagonism were not recorded for any combinations.

## 3. Discussion

The library of 23 benzoxazoles featuring 2-mercaptobenzoxazole with the phenacyl moiety or respective alcohols was screened for the anti-*Candida* activity (Table 2 and Table 3, Appendix A). The active derivatives **5d**, **5i**, **5k** and **6a** (MIC_T/P_ = 16 µg/mL) were tested against the *C. albicans* clinical isolates (resistant to Flu and Itr) and *C. glabrata* displaying an intrinsically decreased susceptibility to azoles [33]. 2,4,6-Trichlorophenacyl- (**5k**) or 5-bromobenzoxazole (**6a**) derivatives showed modest activity (spectrophotometric SPE end-point of %R = 64.2 ± 10.6 and %R = 88.0 ± 9.7 at 16 µg/mL) against the *C. albicans* clinical strain, respectively. The 2,3,4-trichlorophenacyl derivative (**5i**) exerted the most potent anti-*C*. *glabrata* action (%R = 53.0 ± 3.5 at 16 µg/mL). In spite of the total MIC_T_ = 16 µg/mL (%R = 100.3 ± 3.2) against the reference strain, **5d** functionalized with the 4-bromophenacyl group displaying low activity against both azole-resistant isolates (Table 3). The structure activity relationship clearly indicated that the bromo- or chloro-substituted 2-mercaptobenzoxazole with the phenacyl moiety showed activity against the *Candida* reference and clinical azole-resistant strains. Contrary to the literature data [34], the fluoro derivatives were not among the most active compounds. It was evident from the results (Table 4 and Appendix A) that all benzoxazoles at 128 µg/mL displayed more than 81.6% ± 1.8 of the Vero cell viability. The MRC-5 cells showed a viability of app. ≥18.9% when treated with **8h** or **8c**–**d**, **8i** at 512 µg/mL, as well as a ≥ 50.0% cell viability when treated with **5a**, **5e** or **5j**.

We tested the capacity of benzoxazoles to inhibit the ergosterol synthesis (Table 5 and Table 6). The results showed **5d**’s conc. dependent decrease in the ergosterol content when the *C. albicans* ref. strain was grown, respectively, at conc. of 16 or 4 µg/mL. In comparison to the untreated control, the mean of the total ergosterol content decreased in the **5d**-treated *C. albicans* ref. (1.3-fold at 16 µg/mL and 1.2-fold at 4 µg/mL). The above was consistent with the cell proliferation inhibition data (in Table 2, Appendix A), where MIC’s **5d** against SC5314 was the total and %R = 100 was noted. Contrariwise, **5d** displayed the reverse action mode against the *C. albicans* isolate, e.g., the ergosterol content increase of app. 1.3-fold at 16 µg/mL and 1.7-fold at 4 µg/mL vs. the untreated control (Table 5 and Appendix A). Derivative **5d** displayed weak anti-proliferative activity against the following isolates: the *C. albicans* resistant to azoles and *C. glabrata*. HPLC analyses showed that **5d** at 16 µg/mL generated the one-fold increase in ergosterol vs. the untreated control, while the *C. albicans* isolate showed a decrease in ergosterol in line with the increasing concentration of compounds (Table 6).

Exposure to **6a** or **7a** generated a high ergosterol content at 16 µg/mL with the mean decrease in the total ergosterol content of app. one-fold (**6a** or **7a**) for the ref. strain vs. the untreated control. The **6a**- or **7a**-treated clinical isolate displayed a mean decrease of one-fold (**6a** or **7a**) compared to the untreated control (Table 5). Our SPE results of the endogenous sterol quantification (sterols decreased app. 1.2-fold at 16 µg/mL) were consistent with the inhibition of cell proliferation data (SPE reading of end-point % = 100 in Table 2), where **5d** = MIC_T_ against the SC5314 ref and the cell growth reduction (%R = 100) were noted. Additionally, a close agreement between the sterol quantification and broth microdilution results was seen previously by Arthington-Skaggs et al. [35]. Our HPLC results (Table 6 and Appendix A) of the increased endogenous ergosterol in the clinical cells treated with **5d** at 16 µg/mL corresponded with Arthington-Skaggs’s et al. [35] studies on azoles inducing the trailing growth phenomenon. In line with the results of Suchodolski et al. [36] for the strain displaying an uninterrupted ergosterol synthesis, we showed the highest ergosterol accumulation of an app. 0.9-fold increase in the 16 µg/mL-treated ref. strain vs. the untreated control. Since ergosterol was a target for azoles, we showed that the ergosterol content was unsettled in the benzoxazole-treated ref. cells compared to the untreated control (Table 5 and Table 6). While the way of ergosterol depletion in the **5d**-treated *Candida* still has to be elucidated, we clearly demonstrated that the trailing growth phenomenon classified **5d** as active against the blastoconidial cells. Interestingly, the outcomes of our work point to the plasma membrane as a key player in the benzoxazoles’ anti-*Candida* action (Figure 1, Figure 2 and Figure 3 and Appendix A). In compliance with the studies of [37], showing that the *ERG* genes catalysing the late step in ergosterol synthesis were upregulated when the sterol levels were reduced, we concluded that benzoxazoles at MIC_T_ (**5d**) and MIC_P_ (**6a** and **7a**) were subjected to the negative feedback regulation of the ergosterol biosynthesis [37]. We showed that the fungal cell wall is not a barrier to the action mode of benzoxazoles-inducing pyroptosis of the cells and protoplasts at a comparable percentage. Analysing ACD, no changes in susceptibility were noted for benzoxazoles that acted in spite of the cell wall presence (Figure 1) The benzoxazole mechanism was related to the weakened plasma membrane which may have lysed in the presence of MIC and sub-MIC conc. It was shown that the most selected active benzoxazoles induced ACD (Figure 1). Moreover, the tested compounds altered the membrane transport. As we showed (Figure 2), the accumulation of the lipophilic molecules, such as the Rho123 dye, was reduced in the *C. albicans* strains. Thus, when the benzoxazole (**5j** or **8h**) treatment was conducted, the lipophilic molecule, such as the Rho123 tracker, was reduced in the uptake and/or retention in blastoconidials. It was noted that the total conc. of Rho123 in the loading medium was higher (from 1- to 3.3-fold) than within the cells when a lower conc. of **5a**, **5e**, **5j** and **8h** was tested. Our observation demonstrated that mitochondrial respiration was affected/inhibited by the benzoxazoles **8c**–**d** and **8i**. Although, all benzoxazoles exhibited a comparably weak fungistatic effect, they showed diversity in the cell efflux inhibition. The disparity was due to the sequestration of Rho123, leading to the heterogenous sub-cellular localization [38]. The degree of intracellular sequestration was compound-dependent with **8c**–**d** and **8i** at a higher conc., showing a stronger sequestration of ~3.3-fold vs. the lower conc. Therefore, the derivatives substituted with Cl or Br at C-4 or trisubstituted with Cl at C-2,3,4 blocked the fluorophore dye diffuse out of the sequestrating body in the blastoconidial cells. Thus, benzoxazoles could be the uptake transporter of ABCD1′s substrates [39]. Thus, in line with the latter authors [39], we showed that benzoxazoles could inhibit both the passive (micelle-mediated passive diffusion) and the active process occurring via ABC transporters. Moreover, benzoxazoles caused a lethal irreversibility starting with the collapse of the membrane transport through the formation of a leak (Figure 1). We monitored the cell membrane integrity using a PI transported passively into the blastoconidia. We quantified the loss of survival using flow cytometry and determining PI positivity in the benzoxazole-treated cells (Figure 1). Benzoxazole **5j** appeared as the most effective compound inducing an efflux (C_efflux_ = 16 μg/mL or 1.25 μg/mL). This could have be expected, when *CDR1* was overexpressed in *Candida* [40]. Derivative **8h** at C_efflux =_ 1.25 μg/mL was also the most potent inducer of the Rho123 efflux in *C. albicans*, although this value was app. by 4.6-fold higher than at 0.125 μg/mL.

The interaction with ergosterol and MIC showed that there was a specific interaction of **8d** or **8f** with ergosterol, which led to the inhibition of the cell growth (Figure 3). A PI leakage assay (Figure 1) employed in our study supported the hypothesis that benzoxazoles form pores in the fungal membrane similarly to AmB [40]. Since Itr inhibited the ergosterol synthesis and it did not form pores, the latter action mode was the case for **8d** and **8f**. Based on our results of the ergosterol interactions and Rho123 efflux, benzoxazoles (**8d** and **8f**) induced a leakage in ergosterol even at a very high concentration (1200 µg/mL in Figure 3). This indicates that benzoxazoles acted via the membrane barrier perturbation. Thus, benzoxazoles killed yeasts by permeabilizing the plasma membrane (Figure 1). Moreover, they had a similar mode of action as that of Itr inhibiting the ergosterol synthesis [41].

Based on our scaffold screening, derivatives substituted with Br and Cl facilitated the interaction with the negatively charged fungal membrane. They showed the ability to form an amphipathic secondary structure that permitted the incorporation into the outer leaflet of the bilayer of the membrane [42]. In line with Maurya et al. [42], we concluded that benzoxazoles showed a specific antifungal mechanism: an increased permeability and alterations in the cytoplasmic membrane (especially in ergosterol) without the mammalian cell toxicity (lack of ergosterol). Since the sterole-rich membrane was involved in the endo-, exocytosis and efflux, the mode of action of benzoxazoles was related to the important function of ergosterol in yeasts. Benzoxazoles inhibited these important processes by binding to ergosterol; thus, sterols could not perform their functional effects [43]. In line with Escalante et al. [44] and Virág et al. [45], during the interaction of **8d** and **8f** with ergosterol, we considered the possible formation of complexes between the carboxyl group of ergosterol and the OH groups of benzoxazoles (dative bounds) as well as hydrogen bounds. The activity of **5a**, **5j**, **8i** and **8c** (Appendix A) remained unchanged in the presence of exogenous ergosterol, suggesting that they did not interact with the membrane ergosterol.

In general, benzoxazoles can be considered as effective, safe and stable and could be used alone or in a combination therapy with AmB to treat candidiasis (including azole-resistant strains). Since the azoles’ efficiency is limited by the development of *Candida* resistance, we introduced the benzoxazoles active against the clinical isolates resistant to azoles. We selected benzoxazoles reducing efflux pumps; thus, the increase in the number of compounds taken up by the blastoconidia resulted from changes in the ergosterol cell membrane. The intracellular accumulation of benzoxazoles was dependent on an inactive efflux of compound out of the cells. Clearly, further work needs to be undertaken to unravel benzoxazole’s influence on the expression of the ATP-binding cassette (ABC) encoded by the CDR transporters and the major facilitator (MF) encoded by *MDR1*. Benzoxazoles based on their ergosterol biosynthesis-inhibiting capacity, membrane permeabilization activity and increased activity against the azole-resistant isolates, could be an interesting approach to address the problem of resistance. To conclude, the antifungal action of benzoxazoles and the lack of cytotoxicity as well as physico-chemical properties, e.g., good stability, water solubility, rendered **5d** a potential candidate for the development of new antifungal drugs.

## 4. Materials and Methods

### 4.1. Synthesis of Benzoxazoles

Commercially available reagents from Sigma-Aldrich, Fluka and Avantor were used as supplied. Thin-layer chromatography was carried out on TLC aluminium plates with silica gel Kieselgel 60 F_254_ (Merck, Darmstadt, Germany) (0.2 mm thickness film). The column chromatography was performed using Silica gel 60 (Merck, Darmstadt, Germany) of 40–63 μm.

The melting points were measured with a commercial apparatus Thomas Hoover “UNI-MELT” (Thomas, Woonsocket, RI, USA) on samples placed in glass capillary tubes and were not corrected. Infrared (IR) spectra were taken on a Carl Zeiss SPECORD M80 instrument (Carl Zeiss, Jena, Germany).

The ^1^H and ^13^C NMR spectra were measured with a Varian 500 spectrometer (Varian, Palo Alto, CA, USA) operating at 500 MHz for ^1^H and 125 MHz for ^13^C nuclei. Chemical shifts (δ) were given in parts per million (ppm) relative to the residual solvent signal (CDCl_3_, δ_H_ of residual CHCl_3_ 7.26 ppm); signal multiplicity assignment: s, singlet; br s, broad singlet; d, doublet; dd, doublet of doublets; ddd, doublet of doublets of doublets; m, multiplet. Coupling constant (*J*) was given in hertz (Hz). All these measurements were conducted in Warsaw University of Technology.

High-resolution mass spectrometry (HRMS) was carried out on Q Exactive Hybrid Quadrupole-Orbitrap Mass Spectrometer, ESI (electrospray) (Thermo Fisher Scientific, Bremen, Germany) with spray voltage 4.00 kV at IBB PAS Warsaw. The most intensive signals were reported. Ketones 5–7 were synthesized according to procedures reported previously [21,22].

### 4.2. Synthesis of 2-amino-4-bromophenol

To a stirred suspension of 4-bromo-2-nitrophenol (23 mmol, 5 g) in H_2_O (30 mL) and THF (15 mL), activated carbon (9.2 g) was added. The mixture was heated to 50–60 °C, NaBH_4_ (180 mmol, 6.81 g) was added carefully in portions keeping the temperature within the range of 50–70 °C (ca 2–3 h). The reaction was carried out in 60–70 °C (typically 0.5–2 h) and monitored by TLC (hexane/acetate 9/1 *v*/*v*). After full conversion of bromonitrophenol, the mixture was cooled, EtOAc (100 mL) was added, stirring was continued overnight, filtered through a pad of celite, washed with EtOAc (100 mL, in portions), phases were separated, the organic phase was dried (MgSO_4_) and evaporated to dryness. 4-Bromo-2-aminophenol was obtained as orange solid (3.76 g, ca 87%) and used without purification for synthesis of 5-bromo-2-mercaptobenzoxazole **2**.

### 4.3. Synthesis of 2-amino-4,6-dibromophenol

According to procedure reported above, 4,6-dibromo-2-nitrophenol (20.2 mmol, 6 g) in suspension with H_2_O (20 mL), THF (10 mL) and activated carbon (8 g) was reduced with NaBH_4_ (160 mmol, 6.05 g). 4,6-Dibromo-2-aminophenol was obtained as orange solid (4.32 g, ca 80%) and used without purification for synthesis of 5,7-dibromo-2-mercaptobenzoxazole **3**.

### 4.4. General Procedure for Synthesis of Ketones ***5–7***

To a stirred solution of 2-mercaptobenzoxazole **1**–**3** (10 mmol) in MeCN (40 mL) K_2_CO_3_ (15 mmol), appropriate amount of phenacyl chloride or bromide (10 mmol) was added. The reaction was carried out at room temperature (20–22 °C) and monitored by TLC (hexane/acetate 9/1 *v*/*v*). After full conversion of substrates (typically 1 h, for **5k** 24 h), the mixture was poured on ice (120 g), the crude product was filtered, washed with H_2_O and dried on air. Ketones **5** were purified by crystallization, **6** and **7** by column chromatography. After full conversion of substrates (typically 1 h, for **5k** 24 h), the mixture was poured on ice (120 g), the crude product was filtered, washed with water and dried on air. Crude ketones **5** were used for synthesis of alcohols **8** and purified by crystallization for biological research. Ketones **6** and **7** were purified by column chromatography (hexane/EtOAc, gradient, 50/0 to 50/5 *v*/*v*).

### 4.5. General Procedure for Synthesis of Alcohols ***8***

To a stirred suspension of ketone 5 (8 mmol) in MeOH (40 mL) cooled to 0–2 °C, NaBH_4_ (12 mmol) was added in portions keeping the temperature below 2–3 °C (ca 0.5 h). The reaction was carried out at 0–2 °C and monitored by TLC (hexane/acetate 5/1 *v*/*v*). After full conversion of ketone **5** (typically 1–3 h), the mixture was poured on ice (120 g). Solid products (**8b**,**c**,**d**,**i**) were filtered and washed with H_2_O. Oily products were extracted with EtOAc (3 × 50 mL), washed with H_2_O (5 × 50 mL), dried (MgSO_4_), filtered and evaporated. Alcohols **8** were purified by crystallization (hexane/EtOAc 10/1 *v*/*v*) or by column chromatography (hexane/EtOAc, gradient, 50/0 to 50/5 *v*/*v*).

### 4.6. Fungal Strains and Culture Conditions

Antifungal activity of benzoxazole derivatives was tested against the reference *Candida albicans* SC5314 strain from American Type Culture Collection (ATCC). Additionally, two clinical isolates were used in biological activity testing: *C. albicans* (fluconazole- and itraconazole-resistant) and *C. glabrata*. The strains were stored at −80 °C in Microbank system (Pro-Lab Diagnostics, Round Rock, TX, USA). Prior to biological examinations, yeasts were cultured overnight in YEPD (Yeast Extract Peptone Dextrose) or in RPMI 1640 (Roswell Park Memorial Institute, NY, Buffalo, USA) at 30 °C with shaking at 100 rpm. Then, cells were harvested at 3,000 rpm for 5 min at 4 °C and washed twice with sterile water. The final inoculum ranged from 1.9 × 10^7^ to 4.0 × 10^8^ CFU/mL.

### 4.7. In Vitro Antifungal Activity

Antifungal activity was assessed using broth microdilution assay according to the reference CLSI (Clinical and Laboratory Standards Institute) M27-A3 method [13,28,46]. Stock solution of compounds (1600 μg/mL) was prepared in 96% DMSO. Initial inoculum was 10^5^-fold diluted in sterile water and then 1:20 in a medium. Sets of compound test wells (CTW) and sterility control wells (SCW) were prepared in 96-well plates in triplicate for each conc. containing the tested compounds and inoculum (CTW) or sterile water instead of inoculum (SCW) in YEPD or RPMI 1640 medium. Growth control wells (GCW) with inoculum and DMSO (the same amount as in the CTW/SCW wells) in suitable medium were prepared. Plates were incubated for 18 h at 30 °C and evaluated using Synergy H4 Hybrid Reader (BioTek Instruments, Winooski, VT, USA) or with Spark Control M10 (Tecan Group, Männedorf, Switzerland) at 405 nm. Percentage of cell growth inhibition was calculated as follows: % of inhibition = 100 × (1 − (OD_CTW_ − OD_SCW_)/(OD_GCW_ − OD_SCW_). MIC (Minimum Inhibitory Concentration) was determined as the lowest concentration in which a prominent decrease in growth was visible according to the standard criteria from CLSI [28].

### 4.8. Cytotoxicity Assay In Vitro

Cytotoxicity of **5d, 6a** and **7a** was assessed against mammal Vero cell line (ATCC CCL-81, LGC Standards, Poland) using MTS test (3-(4.5-dimethylthiazol-2-yl)-5-(3-carboxymethoxyphenyl)-2-(4-sulfophenyl)-2H-tetrazolium, MTS, Promega, USA) and for **5a**, **5e**, **5j**, **8c**, **8d**, **8h**, and **8i** against mammal MRC-5 cell line (ATCC CCL-171) by MTT assay (3-(4.5-dimethylthiazol-2-yl)-2.5-diphenyltetrazolium bromide, MTT, Thermo Fisher Scientific, USA) [47,48]. Vero and MRC-5 cell lines were maintained in EMEM (Eagle’s Minimum Essential Medium (Merck SA, Darmstadt, Germany) supplemented with 10% FBS (foetal bovine serum, Gibco, Thermo Fisher Scientific, Dublin, Irlandia) and 1% antibiotics at 37 °C and 5% CO_2_. Cells were incubated in 96-well microtiter plates for 24 h. Vero and MRC-5 cell monolayers were treated with different conc. of the benzoxazoles. Following 10 μL of MTS or MTT, reagents were added to each well and then each plate was incubated for 3 h in darkness.

The optical densities were measured at 490 nm (for MTS assay) and 495 nm (for MTT assay) with 660 nm as the reference wavelength using Synergy H4 Hybrid Reader (BioTek Instruments, Winooski, VT, USA) or Spark Control M10 (Tecan Group, Männedorf, Switzerland). Specific absorbance (SA) was calculated using the following formula: SA = A_490/495_ − A_660_. The cytotoxicity was calculated using the following formula: % cytotoxicity = (SA Positive control − SA Test)/(SA Positive control − SA Blank) × 100. Positive control means Vero or MCR-5 monolayer in EMEM and Blank means EMEM with tested conc. of compounds [31].

### 4.9. Ergosterol Estimation Assay Using Spectrophotometry and HPLC

Extraction of ergosterol was performed as previously described in literature with slight modifications [49]. Samples containing 0.5 mL of initial inoculum and tested compound (or no compound in growth controls) in YEPD (final volume 5 mL) were incubated for 24 h at 30 °C with shaking at 100 rpm. Cells were harvested using 3000 rpm for 5 min at 4 °C and washed with sterile water. The wet weight of the cell pellet was determined at the range of 0.04–0.09 g. Then, saponification was performed as follows: 3 mL of 25% alcoholic potassium hydroxide solution was added to each sample and mixed. Cell suspensions were incubated for 1 h at 85 °C in a water bath. Samples were then left to cool at room temp. Ergosterol was extracted by addition of 1 mL of sterile water and 3 mL of petroleum ether followed by vortex mixing. Organic layers were transferred to different tubes. For spectrophotometric analysis, organic layers were diluted five times with ethanol and then analysed using scanned spectrophotometry between 230 and 300 nm with Synergy H4 Hybrid Reader (BioTek Instruments, Winooski, VT, USA). Ergosterol content was calculated using the sum of ergosterol and 24(28)-dehydroergosterol (DHE. late sterol intermediate) according to the formulas [49,50]: %ergosterol + %DHE = (A_282_ × 5/290)/pellet weight (g); %DHE = (A_230_ × 5/518)/pellet weight (g); %ergosterol = (%ergosterol + %DHE) − %DHE.

For quantification of ergosterol by HPLC, organic solvent was evaporated. Sterols were re-dissolved in methanol (HPLC grade) and stored at 4 °C. HPLC analysis were performed using 200 Series Nelson NCI 900 system with UV/VIS detection at 282 nm (PerkinElmer, Waltham, MA, USA) and COSMOSIL C18-MS-II column (Nacalai Tesque, Kyoto, Japan). Gradient elusion was performed starting with acetonitrile: methanol 97:3 *v*/*v* as the mobile phase.

### 4.10. Cytometric Analysis of C. albicans Cell Death Type

Protoplasts of *C. albicans* SC5314 were prepared following the method described in previous article [51]. Cells and protoplasts were treated with **5d**, **6a** and **7a** at conc. of 16 or 4 μg/mL with shaking at 200 rpm at 30 °C for 24 h and cells were treated with the following derivatives: **5e**, **5a**, **5j**, **8h**, **8c**, **8i**, **8d** and **8f** at 16 μg/mL for 24 h. After centrifugation at 3000 rpm at 4 °C for 5 min, the cells were washed and resuspended in sterile water. Staining was preformed according to the annexin and propidium iodide staining kit’s procedure (FITC Annexin V/Dead Cell Apoptosis Kit with FITC annexin V and PI for flow cytometry (Invitrogen, Thermo Fisher Scientific, Dublin, Irlandia) [52]. Suspensions were 10 times diluted in suitable buffer (Invitrogen, Thermo Fisher Scientific, Dublin, Irlandia). Then, 1 μL of annexin was added, mixed and after 15 min of incubation on ice, the cells and protoplasts were centrifuged at 6000 rpm and 4 °C for 5 min and resuspended in buffer. Then, 1 μL of PI was added and after 15 min incubation in ice, fluorescence was measured by flow cytometry with BD FACSLyric 2L6C (BD Biosciences, Franklin Lakes, NJ, USA) for the compounds **5d**, **6a** and **7a,** and with FACSCanto II (Becton Dickinson, National Institute of Drugs, Warsaw, Poland) for **5e**, **5a**, **5j**, **8h**, **8c**, **8i**, **8d** and **8f**. Data were analysed with FACSuite Software 1.4 RUO (BD Biosciences, Franklin Lakes, NJ, USA).

### 4.11. Examination of Rhodamine 123 Efflux in the C. albicans SC5314 Treated with Benzoxazole Derivatives

Procedure was based on the protocol in Gbelska’s article [53,54]. *C. albicans* SC5314 cells were added to YEPD medium and incubated at 28 °C for 20 h with shaking at 150 rpm. Cells were then washed with water and PBS/NaOH. Two microliters of 2-deoxy-D-glucose and Rhodamine123 (Rho123) both at 20 mM were added to 1 mL of the cell suspended in PBS for each tested compound (**5e**, **5a**, **5j**, **8h**, **8c**, **8i**, **8d**). After 18 h incubation at 28 °C and 150 rpm, suspensions were centrifuged and cells were washed with water and PBS. Tested compounds were added to final conc. Of 0.125 or 1.25 μg/mL (**5j**, **8h**, **8i**) and 16 or 160 μg/mL (**5e**, **5a**, **8c**, **8d**). Then, 200 μL of 20 mM glucose were then added to cells and after 18 h incubation at 28 °C with shaking at 150 rpm, 100 μL of supernatant was placed into 96-well plate. Fluorescence of Rho123 was examined with excitation at 521 nm and emission at 600 nm using plate reader Spark Control M10 (Tecan Group, Männedorf, Switzerland).

### 4.12. Antifungal Studies Using Exogenous Ergosterol

To test compounds’ affinity to ergosterol, their antifungal activities in the presence of exogenous ergosterol were examined. Ergosterol (Sigma-Aldrich, Saint Louis, MO, USA) was dissolved in 96% DMSO and Triton X-100 or Tween 20 (to final volumes of 10% and 1%, respectively) [55]. Emulsion was heated, mixed and added to the medium. A 96-well plate was prepared analogically to the reference broth microdilution CLSI M27-A3 method [28]. Compound test wells (CTW) containing C. albicans SC5314 inoculum and separately the following compounds: **5a**, **5e**, **5j**, **8c**, **8d**, **8f**, **8h**, **8i**, exogenous ergosterol in YPD or RPMI 1640 medium were prepared in triplicates. Sterility control wells (SCW) contained tested compounds and ergosterol in medium. The growth control wells (GCW) contained inoculum, DMSO and ergosterol suspended in medium. Final concentration of exogenous ergosterol in each well was 400 μg/mL. Additionally, for examination antifungal activity of **5a, 5e, 5j, 8c, 8d, 8f, 8h, 8i,** wells containing inoculum, ergosterol and itraconazole ITR (Sigma-Aldrich, Saint Louis, MO, USA) in medium (instead of tested derivatives) were prepared. Plates were incubated at 30 °C for 18–96 h and analysed using Synergy H4 Hybrid Reader (BioTek Instruments, Winooski, VT, USA) or with Spark Control M10 (Tecan Group, Männedorf, Switzerland) at 400 or 405 nm.

### 4.13. Determination of Antifungal Effect of Benzoxazole Derivatives Combined with Amphotericin B (AmB). FIC Index Calculation

To determine Fractional Inhibitory Concentration (FIC) index between **5d** and AmB, a modification of checkerboard microdilution method was performed analogically to the CLSI M27-A3 method described above [28]. Compound test wells (CTW) contained as follows: AmB (250 µg/mL; Sigma-Aldrich. USA) (conc. ranging from 2.5 μg/mL to 0.0049 μg/mL) and **5d** (16–0.25 μg/mL) or combination of these two compounds. GCW and SCW wells were prepared as described above in CLSI [28]. Samples were placed in 96-well plate and incubated at 30 °C for 96 h. Growth inhibition rate was measured at 405 nm and percentage of cell growth inhibition in each well was calculated. The FIC index was determined using the formula as follows: FIC = C(**5d**)/MIC(**5d**) + C(AmB)/MIC(AmB), where MIC of **5d** or AmB and **C** stands for conc. of **5d** or AmB in well containing combination of two compounds resulting in at least 50% growth inhibition in a given well in the 96-well microtiter plate in spectrophotometric (SPE) method [56]. Interactions were categorized according to the method of [32] by the following rules: synergism (FIC, ≤0.5), additivity (FIC, >0.5 to ≤1) and indifference (FIC, >1 to <4).

### 4.14. Statistical Analysis

All experiments were performed in triplicate. Data were represented as mean ± SD. Statistically significant differences (*p* values ≤0.05) between the control and test values were determined by means of the Wilcoxon-signed rank-matched pair test using SPSS 16.0 software (Spss 16.0 free download). Moreover, the method comparison studies were determined by the means of regression and Pearson correlation coefficient.

## Data Availability

Not applicable.

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
