# Peer review of "In Vitro Anti-Candida Activity and Action Mode of Benzoxazole Derivatives"

_molecules, 2021, doi:10.3390/molecules26165008_

Round 1
Reviewer 1 Report
The topic of the article is current and interesting. The authors report the obtaining of benzoxazole derivatives some of which are new and have examined theirs in vitro anti-Candida activity. The methods used for the synthesis are well known. The obtained compounds are interesting regarding their biological evaluation. So far, since the beginning of the year, as well as in recent years, interest in this class of compounds is growing. From this point of view, I can conclude that this article will be interesting for the readers in the scientific community.
In general, the text of the whole article should be checked for spelling, grammar, and punctuation errors. There are many of them. Please revise all text very carefully. Make sure to check your English language too.
The abstract does not follow the journal recommendation of a maximum of 200 words. Please reduce the abstract in order to meet the requirements of Molecules.
More keywords may be added – for example: in vitro …
At point 4.1. Synthesis of bezoxazoles – the letter n is missing. This error is all over the text.
Also, at the same point 4.1. in the text, it says that for the synthesis of compounds 2 and 3 is used procedure mentioned in reference 36 with modification. Please explain in the text what is the modification.
Table 9 to be renamed. For example Yields of ketones 5, 6, 7, and alcohols 8.
Information about all the reagents used (supplier) for the synthesis and characterization is to be included in the manuscript and at the beginning in the supplementary file. The authors used melting point, 1H NMR, 13C NMR, IR, and HRMS in order to characterize the newly obtained compounds, so please include all the apparatus models and where the experiments have been made.
For example:
All reagents and chemicals were purchased from commercial sources ….. and used as received. Melting points were determined on a …….. and are un / corrected. The NMR spectral data were recorded on a ………….. spectrometer (Institution—Town, Country). 1H-NMR and 13C-NMR spectra were taken in solvent at …. MHz and at …. MHz, respectively. Chemical shifts are given in relative ppm and were referenced to …………….. (δ = 0.00 ppm) as an internal standard; the coupling constants are indicated in Hz. The NMR spectra were recorded at …. temperature. Mass analyses were carried out on a …………….. mass spectrometer (Make, Town, Country). TLC was carried out on precoated ……. plates (Make, Town, Country).
Please unify the HRMS or HR-MS in the text. We recommend using HRMS abbreviations everywhere in the text. In the supplementary file please calculate and include the mass errors.
Please make the references to the required Molecules style. Use the template provided online.
For example:
Author 1, A.B.; Author 2, C.D. Title of the article. Abbreviated Journal Name Year, Volume, page range
Also, the authors are analyzing the ergosterol content with two different methods – spectrophotometric and HPLC. As you are using two methods for the analysis of ergosterol it’s desirable to give a statistical evaluation in order to indicate which method is more accurate or more appropriate.
In this regard could you comment on which of those two methods is more accurate?
Also, in the text (Table 7) for the content of ergosterol, you are using mg/ml dimension, while in the table in the supplementary file in tables 8s and 9s the ergosterol content is given in %.
Using the same method for ergosterol analysis it’s better if you use the same dimension – for example mg/ml. Perhaps then, it will be possible to statistically assess which method is more accurate.
I recommend the article be published after major revision.
Author Response
Monika Staniszewska Warsaw, August 8, 2021
Corresponding author
Dear Reviewer,
Thank you for all comments included in the letter from the Editor and Reviewers. The correction according to the comments was included in the revised manuscript. The grammatical errors were corrected by a native English speaker.
Authors’ answers to the Reviewer’s comments are included below in the original letter form the Editor. In the case of the Reviewer’s comments (no. 1 and 2) Authors provided the suitable rebuttal in the Table, please see below. Author’s answers are indicated in red and marked in yellow (english corrections), in blue (changes according to the Reviewers’ comments), and in green (the additional Authors’ corrections).
Sincerely yours,
Monika Staniszewska

Reviewer 2 Report
In this submitted manuscript, the authors tested, including half of the known and commercially available benzoxazole derivates for their antiparasitic activity, and further designed a series of experiments to illuminate the possible drug action mechanism. Unfortunately, the paper itself is a real challenge to comprehend, and authors make minimal effort to help.
First of all, there are no structural figures until the very end of the paper, and no feature, such as bromo-substituted or alcohol analogs mentioned during the discussion of the biological activity for the compounds, which combines with random targets design makes it impossible for any meaningful deduction of Structure-Activity Relationship (SAR) or guidance for the future drug design. Second, the table or the figure is presented in a way as tedious and confusing. Despite very reading-unfriendly tables/figures, the logic behind those data collection was largely missing and flawed. For example, no explanation was given why switch the cell-line for different compounds in the cytotoxicity experiment; The confusing red spots in figure 3 and the exact meaning of “binding of benzoxazoles to ergosterol.” Was it a covalent bond or intermolecular interaction between those two? Table 8 needs a serious reorganization and explanation for it to redeem any value. The unit (ug/mL) used in the paper could somehow mislead the biological effectiveness. For instance, when you compare with control compound amphotericin B (AmB), the activity for target compounds seems in a similar range in ug/mL level. Still, the molecular weight of AmB is about four times bigger, which means four times potency to the target compounds in mole concentration. Overall, this paper manages the massive data and experiments poorly, render the rationale and conclusion unconvincing. To be considered for publishing, a serious rewrite is needed.
Author Response
Monika Staniszewska Warsaw, August 8, 2021
Corresponding author
Dear Reviewer,
Thank you for all comments included in the letter from the Editor and Reviewer. The correction according to the comments was included in the revised manuscript. The grammatical errors were corrected by a native English speaker.
Authors’ answers to the Reviewer’s comments are included below in the original letter form the Editor. In the case of the Reviewer’s comments (no. 1 and 2) Authors provided the suitable rebuttal in the Table, please see below. Author’s answers are indicated in red and marked in yellow (english corrections), in blue (changes according to the Reviewers’ comments), and in green (the additional Authors’ corrections).
Sincerely yours,
Monika Staniszewska

Round 2
Reviewer 1 Report
The authors have taken into account all my recommendations.
I think now their manuscript looks much better presented and can be published in present form.